# Predictors of Suicide Re-Attempt in a Spanish Adolescent Population after 12 Months’ Follow-Up

**DOI:** 10.3390/ijerph19137566

**Published:** 2022-06-21

**Authors:** Xavier Alvarez-Subiela, Carmina Castellano-Tejedor, Mireia Verge-Muñoz, Kike Esnaola-Letemendia, Diego Palao-Vidal, Francisco Villar-Cabeza

**Affiliations:** 1Suicide Conduct Unit, Psychiatry and Psychology Department, Sant Joan de Déu Hospital, 08950 Esplugues del Llobregat, Spain; javier.alvarezs@sjd.es (X.A.-S.); mireia.vm15@gmail.com (M.V.-M.); kikesnaola@gmail.com (K.E.-L.); francisco.villar@sjd.es (F.V.-C.); 2Doctoral Program in Psychiatry, Department of Psychiatry and Forensic Medicine, Autonomous University of Barcelona, 08193 Bellaterra, Barcelona, Spain; 3Research Group on Stress and Health (GIES), Department of Basic Psychology, Faculty of Psychology, Autonomous University of Barcelona, 08193 Bellaterra, Barcelona, Spain; carmina.castellano.t@gmail.com; 4RE-FIT Research Group, Parc Sanitari Pere Virgili & Vall d’Hebron Institute of Research, 08023 Barcelona, Spain; 5Unitat de Neurociències Traslacional I3PT-INc, University Hospital Parc Taulí, 08208 Sabadell, Spain; 6Centro de Investigación en Red de Salud Mental (CIBERSAM), Carlos III Health Institute, 28029 Madrid, Spain; 7Institut d’Investigació i Innovació Parc Taulí (I3PT), 08208 Sabadell, Barcelona, Spain; 8Department of Mental Health, University Hospital Parc Taulí, 08208 Sabadell, Barcelona, Spain

**Keywords:** adolescent, suicide behavior, risk factors, prevention

## Abstract

Background: This study aims to identify the main predictive factors that allow for the recognition of adolescents with a higher risk of re-attempting suicide. Method: A longitudinal 12-month follow-up design was carried out in a sample of 533 Spanish adolescents between 12 and 17 years old. The data collection period comprised September 2013 to November 2016, including a one-year follow-up after hospital discharge. Results: A statistically significant regression model was obtained to predict suicide re-attempt at 12-months’ follow-up (χ^2^ = 34.843; *p* < 0.001; Nagelkerke R^2^ = 0.105), including personal history of self-injury (OR = 2.721, *p* < 0.001, 95% CI [1.706, 4.340]) and age (OR = 0.541, *p* = 0.009, 95% CI [0.340, 0.860]), correctly classifying 82.6% of the sample. Our results show that having a personal history of self-injury and being younger than 14 years old were predictors of suicide re-attempt during the first year after an adolescent’s first admission to emergency services. Conclusions: Considering these factors could contribute to the design of more tailored and effective interventions to prevent suicidal behavior in adolescents at high risk of re-attempting suicide.

## 1. Introduction

The prevention of suicide should be a priority for public health systems [1] There are 20 attempts for every completed suicide [1], and although some individuals are more vulnerable than others, one in nine young people have attempted suicide at least once during life [2].

Suicidal ideation and self-injurious behaviors are well-established predictors of new suicide attempts in the future and substantial problems with the social and emotional development of a young person beyond adolescence [3], and between 30 and 50% of re-attempts finish with completed suicide [4].

There is some controversy in the literature about poor adjustment in adulthood being related to adolescent suicidal behavior. Some studies have pointed out psychopathology and psychosocial functioning as clear signs that adolescent suicide attempters have protracted and wide-ranging difficulties that persist into adulthood; however, there is a need for more research to better differentiate if these impairments are strictly related to suicide behaviors during adolescence or premorbid own-risk factors [5].

There are also different genetic (gene modulation), neurobiological (hypothalamic-pituitary-adrenal axis, serotoninergic system, neuroplasticity, neuroimmunological markers, metabolic pattern, and differences in neuroimaging), environmental (stress and life events), and psychological aspects (temperament, character and personality traits, and neuropsychological and neurocognitive factors) that could trigger suicidal behavior [6]. However, the joint study of the interaction of all these factors is challenging; thus, so too is the prediction of who will commit suicidal behavior and when. Therefore, there is still the need for a greater understanding of these associations to unravel these joint interactions, which would facilitate better interpretation of suicide risk with a more careful assessment of suicide risk stratification and planning of clinical and treatment interventions [6,7].

Some other studies have identified risk factors for suicide attempts, such as female sex, being younger than 14 years old, previous psychiatric hospitalizations, previous self-injury behaviors, previous suicidal behavior, family history of psychopathology and family difficulties, stressful life events, and previous history of psychopathology [8,9,10,11]). As can be seen, a broad range of factors have been pointed out, but more research is needed to clarify and identify which predictive factors are the strongest.

Regarding risk factors for suicide re-attempts, studies exclusively based on adolescent samples are scarce. Thus, most of them combine population samples of adults and adolescents. Most referred risk factors from these studies are chronic medical conditions, non-affective psychotic disorder, drug abuse (primarily alcohol abuse), stressful life events (especially sexual abuse), family history of suicide, mental disorders, hopelessness, young age (younger than 20 years old), subjective distress, and impaired general level of functioning [12,13,14,15,16]. These data reveal a direct association between the time elapsed from the first suicide attempt and the risk of re-attempt and increased risk of completed suicide.

The Catalonia Suicide Risk Code Programme (CSRC Programme) [10] is a protocol-based program consisting of suicidal behavior healthcare and preventive interventions launched by the Department of Health of the Catalan regional government, consisting of three phases: (1) early detection and an initial screening of suicide risk, (2) specialized care, and (3) a clinical assessment and preventive follow-up. This secondary suicide prevention program provides a systematic approach to follow-up care for at-risk subjects to ensure patient engagement with the healthcare system. The CSRC combines immediate face-to-face specialized care with telephone management. The program includes a clear action pathway designed to shorten the time between the participant’s first contact with the health care system and the delivery of specialized mental care [10]. The CSRC reported 465 episodes of suicidal behavior between the ages of 12 and 17 in 2019, with 15.7% becoming re-attempts and 2.7% of these re-attempts dying from suicide. Repetition of self-harm is common, with 15–25% of adolescents treated at a hospital for an episode of self-harm returning for treatment within 12 months [17]; this shows the importance of identifying risk factors, and of paying special attention in emergency departments, for re-attempt to be addressed. In this sense, the proper assessment of people attending hospital emergency services after a suicide attempt is fundamental to adding to our knowledge and clarifying the true magnitude of suicidal behavior and its potential repetition. Thus, the identification of risk factors for re-attempt should be a priority to prevent suicide in a group of people who, due to previous attempts at suicidal behavior, may be considered as people at high risk of suicide.

Taking these considerations into account, the objective of this study is to identify predictive factors during screening and assessments in a pediatric emergency service setting. This will allow for the differentiation of suicide attempts with an increased risk of re-attempt from those that do not carry this risk, with the ultimate goal of designing and prioritizing tailored interventions for these individuals, understanding that suicide is rarely due to a single cause and requires a range of prevention initiatives and evaluation methods [18].

## 2. Method

### 2.1. Participants

The aim of this study was to identify risk factors for re-attempt, especially during emergency room assessment, to better comprehend suicide risk with the final aim of planning clinical interventions as Orsolini et al., 2020, strongly recommend [6]. With this objective, we made a longitudinal retrospective study assessing individuals who attended the emergency department due to suicide attempts. The study was first carried out in the mental health area of the pediatric emergency service of the pediatric hospital Sant Joan de Déu, attending to approximately 400 new mental health cases per year, of which about 200 are due to suicidal behavior in children and adolescents. This reference hospital has a catchment area of 1,300,000 inhabitants and receives 100,000 annual visits from pediatric cases sent to the emergency service, being between 10 and 19 years old, a population of 855,157 out of the 7,758,615 total population of our region.

Inclusion criteria were: (1) age from 12 to 17 years old (inclusive), and (2) admission to the psychiatric emergency service due to suicide attempt. The exclusion criteria were: (1) legal adulthood (>18 years), (2) cognitive or other neuropsychological deficits that could hinder clinical assessment and/or understanding of the concept of death, (3) denial of the suicidal intentionality of the behavior, presenting non-suicidal self-harm, intoxication, or other similar behaviors with anxiolytic, playful, or other non-suicidal intention, and (4) refusal to participate in the study.

Finally, a total sample of 533 Spanish adolescents aged between 12 and 17 years old was recruited in a period of three years. 

The age distribution was as follows: 12 years old 5.3% (*n* = 28), 13 years old 13.5% (*n* = 72), 14 years old 23.8% (*n* = 127), 15 years old 20.3% (*n* = 108), 16 years old 18.9% (*n* = 101), and 17 years old 18.2% (*n* = 97). We chose to dichotomize between 14 and younger and older than 14 because scientific research indicates that 34.6% of mental disorders have their onset before 14 years of age [19]. Globally, suicide is the second leading cause of death in 15−29 year-olds [1], so we point out the importance of suicide attempts in younger ages. Age was dichotomized (aged 14 years old or younger vs. older than 14) in the regression analyses (χ^2^ = 8.181, df = 1, *p* = 0.004, Eta = 0.124).

The patients’ genders were not considered in the sample data collection. Instead, biological sex (male or female) was registered, since this is how it appeared in the computerized patients’ history at the time of the study.

### 2.2. Instruments

The main outcome variable of the study was the repetition of a suicide attempt.

This variable was evaluated at 12 months of follow-up from the first admission of the subject to the psychiatric emergency service of the referral hospital, requiring a hospital admission where the subject entered the investigation. During the evaluation period, all the subjects seen in the emergency room of our hospital required a minimum pediatric hospital stay of 72 h due to standard protocol requirements.

Demographics (sex, age), clinical data (diagnoses, psychiatric comorbidity, and past personal and familial psychiatric diagnoses, treatment, and evolution), social and past history variables (bullying and sexual abuse), and variables related to suicide attempts (previous, characteristics, type, method, and re-attempt) were collected by clinical interview with the adolescent and the family and by reviewing the computerized medical records. An expert professional (psychiatrist or psychologist) established the psychiatric diagnosis through a clinical interview using DSM-IV TR criteria [20].

We consider suicidal behavior, as Al-Halabí et al. [18] suggest, as being the set of thoughts and behaviors related to intentional suicide, and attempts as engagement in potentially self-destructive behavior in which there is at least some intention to die as a result of the behavior, differentiating this from non-suicidal self-injuries where the final intention has nothing to do with death. Furthermore, re-attempt is considered as being a new attempt during the research period. 

No assessment instruments were used beyond those commonly used in clinical practice because of the naturalistic essence of the research in hospitalization and the emergency room.

### 2.3. Procedure

Data collection was from September 2013 to November 2016 to include subjects at 12-months’ follow-up.

The collection and coding of the data were carried out on the same day as the individual’s emergency admission and following hospital admission. All data were collected by a clinical professional (psychologist or psychiatrist) belonging to the psychiatry service of the Sant Joan de Déu Hospital, with a complete first mental health exploration being coded in the clinical background program used for hospitalizations.

During the study period, after a year of attention in the emergency room, we reviewed the electronic health records and evaluated the variables of those who re-attempt noted by professional therapists (psychologists or psychiatrists), introducing these variables in a database. 

### 2.4. Ethical Considerations

The study complies with the internal regulations of the ethics committee of Sant Joan de Déu Hospital and those of the World Medical Association and the Declaration of Helsinki of 1995 [21], with its successive amendments. Since no additional data were collected and no other invasive procedures were performed on subjects, no additional informed consent was required other than the standard consent provided at the time of emergency admission. After receiving information regarding the study, its objectives, and the agreement of confidentiality and protection of personal data, all participants and families gave their written consent. Participation in this study was not remunerated.

The project was accepted by the CEIm (Ethics Committee) Sant Joan de Déu Foundation with the internal code PIC-158-18.

### 2.5. Data Analysis

All statistical calculations were performed using the Statistical Package for the Social Sciences (SPSS) version 25. Descriptive statistics and frequency distribution analyses of all variables considered in the present study were calculated. Differences between the groups were analyzed with a Student’s t-test and one-way variance analysis (ANOVA) for independent samples (for quantitative variables) or with a chi-squared test (or the Fisher’s exact test when no application criteria were met for the chi-squared) calculated from 2 × 2 contingency tables (for categorical variables). To analyze predictors of the main variable of the study (repetition of suicidal behavior at 12-months’ follow-up), a binary logistic regression analysis was carried out (stepwise method), using those variables that showed a statistically significant relationship with the main variable of this study. The Bonferroni multiple-comparison post-hoc correction was employed. The significance of all tests was considered at a probability level of 5% or less, always indicating the exact significance and confidence interval of 95% offered by the SPSS.

## 3. Results

### 3.1. Description of the Sample

Data from 533 participants were collected over three years (see Table 1), including a one-year follow-up after hospital discharge. The sample was predominantly female (*n* = 446, 83.7%) with an average age of 14.89 years (SD = 1.47, range 12–17).

Focusing on those adolescents with a suicide re-attempt (*n* = 93, 17.4%), 92.5% were females (*n* = 86), with a mean age of 14.54 years old (SD = 1.29, range 12–17; 44.1% > 14 years old).

Significant differences were found between groups (re-attempt vs. no re-attempt) with regard to sex distribution (χ^2^ = 6.381, df = 1, *p* = 0.012, Eta = 0.109), with females displaying higher rates of re-attempt compared to males. Similarly, significant differences were found between groups (re-attempt vs. no re-attempt) with regard to age (t_149.193_ = 2.795, *p* = 0.006, CI 95% 0.723–0.124), with younger adolescents showing higher rates of re-attempt (M = 14.14, SD = 1.3 vs. M = 14.96, SD = 1.5).

### 3.2. Clinical Data

Table 2 displays clinical data related to diagnosed psychopathology of the sample at the time of the interview. Considering the whole sample, 93.2% of participants (*n* = 497) had some psychiatric diagnosis, with this percentage being significantly higher in cases of re-attempted suicide (χ^2^ = 5.769, df = 1, *p* = 0.016, Eta = 0.104). Comorbidities occurred in 31% of cases (*n* = 165), increasing up to 40.9% in adolescents presenting a re-attempt (*n* = 38); this was statistically significant (χ^2^ = 5.169, df = 1, *p* = 0.023, Eta = 0.098).

The most frequently observed diagnoses in the whole sample considering both groups were adjustment disorders (*n* = 233, 43.7%), affective disorders (*n* = 131, 24.6%), and personality disorders (*n* = 116, 21.8%). Significant differences between groups were observed only for personality disorders (χ^2^ = 8.857, df = 1, *p* = 0.003, Eta = 0.129) with higher rates in the suicide re-attempt sample of adolescents.

Table 3 displays clinical data related to personal background, considering personal background as all the clinical circumstances related to psychopathology and other relevant clinical situations that precise evaluation but were not happening during the emergency evaluation. Considering the whole sample, 71.3% of studied adolescents (*n* = 380) revealed personal background related to mental health problems, reaching 81.7% within the re-attempt subgroup (*n* = 76), which is significantly higher (χ^2^ = 5.984, df = 1, *p* = 0.014, Eta = 0.106), therefore, demonstrating that this might be a relevant risk factor for repeated attempts.

The most frequent clinical backgrounds of the overall sample, considering both re-attempt and non-re-attempt subgroups, were non-suicidal self-harm (*n* = 210, 39.4%), previous suicidal behavior (*n* = 196, 36.8%), affective disorders (*n* = 81, 15.2%), and bullying (*n* = 75, 14.1%). Significant differences were found between groups in personal background of self-harm behaviors (χ^2^ = 22.610, df = 1, *p* < 0.001, Eta = 0.206), previous suicidal behavior (χ^2^ = 5.382, df = 1, *p* = 0.020, Eta = 0.1), and bullying (χ^2^= 6.746, df = 1, *p* = 0.009, Eta = 0.113).

Table 4 displays family background and clinically-related data. For the sample as a whole, 56.1% of studied adolescents (*n* = 299) revealed family backgrounds in terms of mental health problems; for the re-attempt group, the figure was 63.4% (*n* = 59). 

For the overall sample, the most frequently observed family complications were affective disorders (*n* = 172, 32.3%), substance use (*n* = 83, 15.6%), and family suicidal behavior (*n* = 68, 12.8%). When comparing family background between groups, no significant differences were found.

When the factors that showed statistically significant relationships with suicide attempt and re-attempt at 12-months’ follow-up were introduced into a binary logistic regression model (stepwise), a statistically significant model was obtained χ^2^(3, *n* = 533) = 34.843; *p* < 0.001; Nagelkerke R^2^ = 0.105, including personal history of non-suicidal self-harm (OR = 2.721, *p* < 0.001, 95% CI [1.706, 4.340]) and age (OR = 0.541, *p* = 0.009, 95% CI [0.340, 0.860]) (see Table 5). This model correctly classified 82.6% of the sample. Therefore, having a personal history of self-injury and being 14 years old or younger were revealed as predictors of suicide re-attempt during the first year after the adolescent’s first admission to emergency services.

## 4. Discussion

The rate of suicide re-attempts in our sample with a 12-months’ follow-up was around 17%, coinciding with 15–20% of rates on average revealed in similar studies [22]. However, studies with larger follow-ups in which re-attempt was defined as any type of suicidal behavior found rates of around 30–40% [23].

Most research on suicide carried out with adolescent populations has shown that higher rates of suicidal behavior are observed at around 14–15 years of age [24,25]. Age has been repeatedly described as a risk factor for repeated suicide attempts [26]. In this sense, a higher probability of suicide attempt repetition is associated with lower age (usually described as under 15 years old) when the first episode has occurred [26]. In our sample, despite higher rates of re-attempt in adolescents aged 14 or younger (55.91%), the mean age for re-attempt was 14.54 years old. 

Adolescents who have a history of non-suicidal self-harm have a substantially increased risk of adverse non-fatal and fatal outcomes, including suicide, compared with those who do not self-harm [3]. It is important to note that in our study, when performing stepwise analysis, previous suicidal behavior did not appear as a significant predictor of re-attempt, but non-suicidal self-harm did. This probably has to do with the fact that, in most of our sample, individuals with a history of prior attempts were the same as those who had previously inflicted self-harm; as self-harm was more frequent, this might have masked the statistical result of the previous suicidal behavior.

Sex did emerge as a factor related to higher rates of re-attempt. However, it did not emerge as a significant predictor when entered into a stepwise binary logistic regression model. The same is true for other variables, such as comorbidity, mental health diagnosis, previous suicidal behavior, and bullying. A few variables, such as sexual abuse and family history of suicide behavior, did not appear as significant in the preliminary statistics. Although it did not emerge as a statistically significant factor related to suicide re-attempt, another important variable of our study was sex. Our sample was predominantly female, and this distribution is consistent with those found in other studies in which most attempts were carried out by young women [27]. Women attempt suicide three times more than men, but men die by suicide three times more than women, with some exceptions, as the article from Al-Halabi and Fonseca-Pedrero (2021) explains. Despite being one of the significant factors in the comparison between subjects with and without re-attempts, it did not turn out to be statistically significant in the regression; we take it that these differences may be due to the studied relationship between non-suicidal self-harm and female sex, as indicated by the meta-analysis conducted by Bresin and Schoenleber in 2015 in which the chances of non-suicidal self-harm were described as significantly higher among females [28].

Jakobsen and collaborators [29] explained that mental health disorders are strong predictors of suicide re-attempts in adolescents and those with comorbid diagnoses are at higher risk. Comorbidity was not revealed as a statistically significant predictive factor for re-attempt in our research despite this evidence.

Several diagnoses have been associated with suicidal behavior, and depressive symptoms are among the most common. However, anxiety, affective disorders, disruptive behavior, and substance disorders were also important variables for suicide behaviors in adolescents [30]. Concerning predisposing factors among mental health disorders, the only ones that showed significant differences between those who re-attempted and those who did not were personality disorders and maladaptive personality traits. This has been described in other studies, both in adolescent and adult sample populations, and it is accepted by the scientific community, especially regarding borderline personality disorder. It is also supposed to be the case with maladaptive personality traits, although there are no compelling data on these symptoms [31].

As stated in previous research, involvement in bullying as a victim and/or perpetrator in both traditional and electronic contexts is associated with increased suicide risk [32]. In our study, even though it seemed to be significant as a single variable, it did not seem to be relevant when we applied a regression analysis. In future studies, we will control for this variable more carefully to be able to reach better conclusions in the near future. We will also attend more carefully to the association between cyberbullying and suicidal behavior seen in other studies [33,34].

Because many investigations of sexual abuse in childhood relate these traumatic experiences to suicidal behavior [35], we have considered it essential to include this factor in this research. Sexual abuse in childhood is common in all societies; the findings from studies to date suggests that approximately 4% of girls and 2% of boys experience childhood sexual assault each year, with the majority occurring in the teenage years [36], with an average prevalence of around 20% in women and 8% in men. In Spain, the prevalence described is between 7 and 20% [37], and in our sample, it reaches 5% (lower than the Spanish average), so this may be why this factor is not revealed as significant in our research.

With regard to a familial history of psychopathology, multiple studies have described its relationship to suicidal behavior in the offspring, involving difficulties in adaptability and family cohesion, difficulties in problem-solving, and negative parental relationships as predictors of suicidal behavior [38,39]. Mental health problems and suicidal behavior in close relatives are commonly related and frequently identified as risk factors for suicide attempts and re-attempts in adolescents [39]. These relationships were not found in our research, which leads us to hypothesize that pain, shame, or the underestimation of its relevance could have inhibited relatives from disclosing such information during data collection in emergency care.

### 4.1. Limitations

A retrospective study design has poor scientific value in comparison with other studies, but it helps us to evidence the need for more studies with adolescents, and to elaborate on more specific projects to properly evaluate the risk factors of suicidal behavior. 

Despite the wide and diverse population belonging to our care region, the data of this research came from a single hospital. Therefore, the results cannot be generalized to the general population in other countries and outside our follow-up nucleus. Neither the financial status of the families nor the ethnicity of the subjects was collected in our database, which also makes it difficult to attribute these results to the general population.

Most of the variables were obtained through clinical interviews or from electronic health records, not using instruments, questionnaires, or psychometric measures, so biases could appear in relevant data to our investigation, such as a history of sexual abuse, bullying, or family background. This is a regular procedure in the hospital where the study was carried out to avoid overloading a population that arrives in a delicate moment and state, and for this reason, only these clinical history data are collected. The aim is to improve decision-making with variables that can be extracted in an emergency visit.

Another limitation is the difficulty in carrying out a follow-up with subjects older than 18, including a possible bias in the relationships found between suicide re-attempt and age, although our data do coincide with previous research of a similar nature.

It is also important to note the lack of information from private clinical practices, even though with our national health system there are not many subjects receiving private care.

Other risk data missing are deaths by suicide during this period. In the Catalonia Suicide Risk Code (CSRC), only two male adolescents appear as exitus during the study period, and therefore are not statistically significant for our study. 

The over-represented female sex in our sample could be another limitation to consider related to statistical power, despite its similarity to other study samples of similar research in this field.

Lastly, we did not take into account the specific dates of the attempts, which could draw seasonal profiles affecting behavior including suicidal behavior, especially for the female sex as suggested in a study by Tonetti in 2007 [40].

### 4.2. Strengths

As already mentioned in the introduction, only a few studies have been carried out on the risk factors of re-attempt exclusively based on the adolescent population. This confers a particular value on the present research. However, we still need to delve more deeply into the data and risk factors, evaluating those not identified in this research but usually described in other studies.

Finally, prospective studies are needed and could lend value to our results, as our research was intended to generate new questions to answer, such as the importance of family relationships and family mental pathology in adolescents with suicidal behavior, the impact of bullying on suicide attempts, the relationship between suicidal behavior and new technologies, and the question of whether treatment therapies for subjects with self-harm also decrease suicidal behavior in these subjects. These are projects for future study.

## 5. Conclusions

Our study traces the characterization and evolution of suicidal behavior and re-attempts in a sample of adolescents, making us aware of the importance of prevention and highlighting two key aspects. The first aspect is the importance of non-suicidal self-harm behavior since it is an easily identifiable indicator of a poor prognosis of subjects with suicidal behavior. We may conclude that as long as self-injurious behavior is present in subjects with suicidal ideation, the risk of a re-attempt is very high. Moreover, this is a clear target on which to focus our efforts for prevention, both in outpatient units with specific treatments, frequent assessments, and group therapies, and in emergency units where we can prioritize a hospital admission or communicate to outpatients’ units to manage these situations. This is most likely because the presence of self-harmful behavior as a strategy to cope with the demands of daily life is an indicator of the lack of the adaptive resources of the individual. The second highlighted aspect is that early onset of suicidal behavior, specifically before the age of 14, was identified as a potential predictor of a worse evolution than in late onset, which may be explained by a continuous increase in interpersonal problems, differences in family care, academic and work-related problems, and social demands that adolescents have to face in their transition to adulthood, being important factors to focus on for prevention in adolescents. This knowledge will allow us to design specific individualized and group therapies to work with higher-risk subjects, taking into account these profiles when they go to the emergency room with suicidal behavior. We may even approach prevention by treating specific groups, such as women with a history of non-suicidal self-harm and those with family mental pathology. Finally, we need to have clear risk factors in order to establish elaborate, good prevention programs and to reduce the frequency of suicidal behavior, even suicide deaths, and more studies are required to reach this goal.

## Figures and Tables

**Table 1 ijerph-19-07566-t001:** Variables of the sample.

Variables	(*n*, %)
Demographics and clinical data	
Sex (female)	(*n* = 446, 83.7%)
Age (older than 14 years old)	(*n* = 306, 57.4%)
Attempt method	Medication overdose (*n* = 336, 63%)
Current diagnosis	
Any psychiatric diagnosis	(*n* = 497, 93.2%)
Psychiatric comorbidity	(*n* = 165, 31%)
Substance use	(*n* = 32, 6%)
Schizophrenia and psychotic symptoms	(*n* = 12, 2.3%)
Affective disorders	(*n* = 131, 24.5%)
Anxiety disorders	(*n* = 31, 5.8%)
Eating disorders	(*n* = 57, 10.7%)
Impulse control disorders	(*n* = 2, 0.4%)
Adjustment disorders	(*n* = 233, 43.7%)
Personality disorders or traits	(*n* = 116, 21.8%)
Behavioral disorders	(*n* = 36, 6.8%)
Autism spectrum disorders	(*n* = 12, 2.3%)
Attention deficit and hyperactivity disorders	(*n* = 12, 2.3%)
Other diagnosis	(*n* = 24, 4.5%)
Personal background of mental disorder	
Existence of personal background	(*n* = 380, 71.3%)
Substance use	(*n* = 28, 5.3%)
Schizophrenia and psychotic symptoms	(*n* = 6, 1.1%)
Affective disorders	(*n* = 81, 15.2%)
Anxiety disorders	(*n* = 67, 12.6%)
Eating disorders	(*n* = 69, 12.9%)
Impulse control disorders	(*n* = 0, 0%)
Adjustment disorders	(*n* = 42, 7.9%)
Personality disorders or traits	(*n* = 27, 5.1%)
Behavioral disorders	(*n* = 57, 10.7%)
Autism spectrum disorders	(*n* = 7, 1.3%)
Attention deficit and hyperactivity disorders	(*n* = 33, 6.2%)
Self-harm	(*n* = 210, 39.4%)
Abuse during childhood	(*n* = 31, 5.8%)
Bullying	(*n* = 75, 14.1%)
Mental health hospitalization	(*n* = 78, 14.6%)
Previous suicide behavior	(*n* = 196, 36.8%)
Other personal background	(*n* = 70, 13.1%)
Family background of mental disorder	
Family background of psychopathology	(*n* = 299, 56.1%)
Substance use disorders	(*n* = 83, 15.6%)
Schizophrenia and psychotic symptoms	(*n* = 44, 8.3%)
Affective disorders	(*n* = 172, 32.3%)
Anxiety disorders	(*n* = 69, 12.9%)
Eating disorders	(*n* = 16, 3%)
Impulse control disorders	(*n* = 0, 0%)
Adjustment disorders	(*n* = 16, 3%)
Personality disorders or traits	(*n* = 20, 3.8%)
Behavioral disorders	(*n* = 18, 3.4%)
Family history of suicidal behavior	(*n* = 68, 12.8%)
Other family background of psychopathology	(*n* = 59, 11.1%)

**Table 2 ijerph-19-07566-t002:** Diagnosed psychopathology of the sample.

Variables	Relapse (*n* = 93)	No Relapse (*n* = 440)	*p*
	*n* (%)	*n* (%)	
Substance use disorders	5 (5.4%)	27 (6.1%)	n.s.
Schizophrenia & psychotic symptoms	1 (1.1%)	11 (2.5%)	n.s.
Affective disorders	25 (26.9%)	106 (24.1%)	n.s.
Anxiety disorders	4 (4.3%)	27 (6.1%)	n.s.
Eating disorders	12 (12.9%)	45 (10.2%)	n.s.
Impulse control disorders	0 (0%)	2 (0.5%)	n.s.
Adjustment disorders	42 (45.2%)	191 (43.4%)	n.s.
Personality disorders or traits	31 (33.3%)	85 (19.3%)	0.003
Behavioral disorders	6 (6.5%)	30 (6.8%)	n.s.
Autism spectrum disorders	4 (4.3%)	8 (1.8%)	n.s.
Attention deficit & hyperactivity disorders	2 (2.2%)	10 (2.3%)	n.s.

Note. n.s.: non-significant differences between groups according to the chi-square test.

**Table 3 ijerph-19-07566-t003:** Clinical background of the sample.

Variables	Relapse (*n* = 93)	No Relapse (*n* = 440)	*p*
	*n* (%)	*n* (%)	
Substance use disorders	3 (3.2%)	25 (5.7%)	n.s.
Schizophrenia & psychotic symptoms	1 (1.1%)	5 (1.1%)	n.s.
Affective disorders	17 (18.3%)	64 (14.5%)	n.s.
Anxiety disorders	15 (16.1%)	52 (11.8%)	n.s.
Eating disorders	17 (18.3%)	52 (11.8%)	n.s.
Impulse control disorders	0 (0%)	0 (0%)	n.s.
Adjustment disorders	8 (8.6%)	34 (7.7%)	n.s.
Personality disorders or traits	3 (3.2%)	24 (5.5%)	n.s.
Behavioral disorders	9 (9.7%)	48 (10.9%)	n.s.
Autism spectrum disorders	2 (2.2%)	5 (1.1%)	n.s.
Attention deficit & hyperactivity disorders	4 (4.3%)	29 (6.6%)	n.s.
Non suicidal self-harm	57 (61.3%)	153 (34.8%)	<.001
Sexual abuse during childhood	4 (4.3%)	27 (6.1%)	n.s.
Bullying	21 (22.6%)	54 (12.3%)	0.009
Mental health hospitalization	16 (17.2%)	62 (14.1%)	n.s.
Previous suicidal behavior	44 (47.3%)	152 (34.5%)	0.020

Note. n.s.: non-significant differences between groups according to the chi-square test.

**Table 4 ijerph-19-07566-t004:** Family background of the sample.

Variables	Relapse (*n* = 93)	No Relapse (*n* = 440)	*p*
	*n* (%)	*n* (%)	
Substance use disorders	20 (21.5%)	63 (14.3%)	n.s.
Schizophrenia & psychotic symptoms	10 (10.8%)	34 (7.7%)	n.s.
Affective disorders	38 (40.9%)	134 (30.5%)	n.s.
Anxiety disorders	14 (15.1%)	55 (12.5%)	n.s.
Eating disorders	2 (2.2%)	14 (3.2%)	n.s.
Impulse control disorders	0 (0%)	0 (0%)	n.s.
Adjustment disorders	4 (4.3%)	12 (2.7%)	n.s.
Personality disorders or traits	4 (4.3%)	16 (3.6%)	n.s.
Behavioral disorders	5 (5.3%)	13 (3%)	n.s.
Autism spectrum disorders	0 (0%)	0 (0%)	n.s.
Attention deficit & hyperactivity disorders	0 (0%)	0 (0%)	n.s.
Family suicidal behavior	16 (17.2%)	52 (11.8%)	n.s.

Note. n.s.: non-significant differences between groups according to the chi-square test.

**Table 5 ijerph-19-07566-t005:** Statistical significant model for binary logistic regression (χ^2^(3, *n* = 533) = 34.843; *p* < 0.001; Nagelkerke R^2^ = 0.105).

Variables	OR	*p*	CI
Non-suicidal self-harm	2.721	<0.001	95% [1.706, 4.340]
Age	0.541	0.009	95% [0.340, 0.860]

## Data Availability

The data that support the findings of this study are available from the corresponding author upon reasonable request.

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
