# Peer review of "Predictors of Suicide Re-Attempt in a Spanish Adolescent Population after 12 Months’ Follow-Up"

_ijerph, 2022, doi:10.3390/ijerph19137566_

Round 1

Reviewer 1 Report

Thank you for allowing me to review this paper. I reviewed it with great interest, as it is current and vital. It deals with critical issues of mental health. As authors mentioned, globally, suicide is the second leading cause of death in 15−29 year-olds, thus I have no doubt about the necessity of those kind of studies.

The construction of the study is apropriate, all of the methods were correctly used and described. I have to underline the applicative aim of the study (provide useful information about predictors of potential suicide re-attempts).

However, I have some specific points to increase the level of the manuscript:

1. Summary needs to be revised. Please, put the sample' information. It does not appear until the line no. 198. Please, do the same with the name of the program (CSRC) there. Besides, it's worth to put the proper citation within the lines 76-79. 

2. To be precised, is the CSRC Programme a base for the whole study? One may have doubts about it.

3. It's worth to add the total size of the sample between the lines no. 121-129.

4. Is it possible to add the information about the total numbers of population, which were taken into account? (lines 114-120) I'm curious about the numbers regarding to point 4. (line119)

5. Paragraph between the lines 48-58  needs a more in-depth bibliographic review. Here one could find a lot of predictors or determinants without bibliografic connotation. 

6. Is the Orsolini's publication the only one to describe the Authors' topic between the lines 104-107?

Author Response

Dear reviewer 1, thanks for taken the time to read and comment on our research. Below, we provide specific response to each question. We hope we have amended all your concerns in this new version of the manuscript.

Point 1: Summary needs to be revised. Please, put the sample' information. It does not appear until the line no. 198. Please, do the same with the name of the program (CSRC) there. Besides, it's worth to put the proper citation within the lines 76-79. 

Response 1:  

Thank you for your useful comments. We add more data about the sample in the summary. Now it reads as follows:

“A longitudinal 12-month follow-up design was carried out  in a sample of 533 Spanish adolescents between 12 to 17 years old. Data collection period comprised September 2013-November 2016, including one year follow-up after hospital discharged”.

We also add the reference for the Catalonia Suicide Risk Code Progamme in the fisrt appearance. The Catalonia Suicide Risk Code Programme is a very valuable and useful tool for tracking and monitoring data base of suicide behavior in clinical patients. It is also very important to deepen in the scientific and clinical knowledge for our territoty, allowing different research designs. Therefore, attending to the reviewer comments, we have included more details about the CSRC in the Introductory section of the manuscript. However since the CSRC is not the base for our research (the sample was not recruited from the CSRC), this was not mentioned in the abstract.

Point 2: To be precised, is the CSRC Programme a base for the whole study? One may have doubts about it.

Response 2:

Thank you again for the commments. We understand your doubt. As stated in our previous response to comment 1, the CSRC is a very useful tool in our clinical reality, allowing a proper and eficient data tracking and monitoring concerning suicide behaviour in our specific region. However, our sample was not recruited or extracted from the CSRC. It was directly recruited form patients admitted to our reference hospital. For this reason, the CSRC is mentioned in the introduction, but not in the methods section. We hope it is clearer now.

Point 3: It's worth to add the total size of the sample between the lines no. 121-129.

Response 3:

This is true. Thanks for noticing. We have inclouded the following sentece in line 121 from the methods section: “Finally, a total sample of 533 Spanish adolescents aged between 12-17 years old was recruited in a period of three years”. 

Point 4: Is it possible to add the information about the total numbers of population, which were taken into account? (lines 114-120) I'm curious about the numbers regarding to point 4. (line119)

Response 4:

The total number of population taken in account is clarified in the revised version of the manuscript. New information reads as follows: “This reference hospital has a catchment area of 1,300,000 inhabitants and receives 100,000 annual visits of pediatric cases to the emergency service, being between 10 to 19 years old population of 855,157 from the 7,758,615 of the total population of our region”

Point 5: Paragraph between the lines 48-58 needs a more in-depth bibliographic review. Here one could find a lot of predictors or determinants without bibliografic connotation. 

Response 5:

Thank you for the comment. We have added the reference of Orsolini et al., 2020 from where is primarily the information we comment on this lines.

“There are also different genetic (gene modulation), neurobiological (hypothalamic-pituitary-adrenal axis, serotoninergic system, neuroplasticity, neuroimmunological markers, metabolic pattern, differences in neuroimaging), environmental (stress life events), and psychological aspects (temperament, character and personality traits, neuropsychological and neurocognitive factors) that could trigger suicidal behavior (Orsolini et al., 2020)”.  

Point 6: Is the Orsolini's publication the only one to describe the Authors' topic between the lines 104-107?

Response 6:

It’s true that this is not the only one. However, for the sake of clarity, we have only included this one since it is a recent reference that agglutinates information on this field from a well-know research in this topic.

Reviewer 2 Report

The manuscript contains data of interest, although the authors should review this version in accordance with my suggestions and concerns detailed below:

- They affirm that they have used the DSM-5 criteria (the reference must be incorporated - and it is better not to use Roman numerals), although the detail of the diagnostic categories provided in the tables raises doubts. In version 5, the category of affective disorders disappears and the main diagnostic considerations are modified with respect to DSM-IV.

- For a considerable improvement of the current version and since the authors have the data, the criterion of month of the year (or season) in which both the first and second attempts were carried out should be incorporated. In many mental disorders there are seasonal variations that affect behavior, including suicidal behavior. There is no doubt about the influence of this not only in countries with large seasonal variations, but also in the Mediterranean ones as from which the data comes. It is suggested to incorporate the pioneering work of Tonetti et al. (2007, Journal of Affective Disorders, 97 (1-3), 155-160) and carry out these additional analyzes also assessing whether this can be a predictor variable in the logistic regression model.

– The discussion should be limited more to strictly necessary elements, part of the contents of previous studies should be incorporated in the introduction. This will help shorten the length of the discussion. In the same way, the conclusions are excessively long, they must be delimited with the essential text (for example, the part developed in the first paragraph of page 11 can be summarized in a single sentence avoiding speculations of prevention).

- The reference section should be checked carefully. There are numerous errors, such as the current references 1-2, 3-4, 5-6, 7-8, 9-10,... in which the second contains only the link to the first. The authors must then assess the number of references the manuscript really has (probably half). Do they correspond to all those included in the text? No numbering has been used in this one. All of these formatting issues need to be adjusted accordingly.

– The age of the participants must be detailed in the abstract and check to always use adolescents. In the current version the last sentence of the abstract uses “young subjects”.

- Finally, in the title it is necessary to specify that the adolescent population is Spanish.

Author Response

Dear reviewer 2, thanks for the time taken to read and comment on our research. Below, we provide specific response to each question. We hope all issues raised have been amended in this new version of the manuscript.

Point 1: They affirm that they have used the DSM-5 criteria (the reference must be incorporated - and it is better not to use Roman numerals), although the detail of the diagnostic categories provided in the tables raises doubts. In version 5, the category of affective disorders disappears and the main diagnostic considerations are modified with respect to DSM-IV.

Response 1:

Thank you for the comment and we are sorry for the mistake. It was used the DSM-IV TR Manual. Thus, this reference (number 3 in the reference list and in page 4 of the text) was added to the manuscript.

Amercian Psychiatric Association (APA). (2002). Manual Diagnóstico y Estadístico de los Trastornos Mentales DSM-IV-TR. Barcelona: Masson.

Point 2: For a considerable improvement of the current version and since the authors have the data, the criterion of month of the year (or season) in which both the first and second attempts were carried out should be incorporated. In many mental disorders there are seasonal variations that affect behavior, including suicidal behavior. There is no doubt about the influence of this not only in countries with large seasonal variations, but also in the Mediterranean ones as from which the data comes. It is suggested to incorporate the pioneering work of Tonetti et al. (2007, Journal of Affective Disorders, 97 (1-3), 155-160) and carry out these additional analyzes also assessing whether this can be a predictor variable in the logistic regression model.

Response 2:

Thanks again for the interesting comment. Despite the potential relevance of this fact raised by the reviewer, we regret to inform that this specific factor (data) was not collected and monitored during the present research. Specific dates of the attempt were not collected not being an objective of the present research. However, it is true that a seasonal effect could not be outlined. For this reason we have decided to add the following sentence in the Limitations’ section of the present manuscript:

“Lastly, it wasn’t taken in account the specific dates of the attempts which could draw seasonal profiles affecting behavior including suicidal behaviors, especially in female gender as suggests the study of Tonetti in 2007 (Tonetti et al., 2007)”.

In addition, we would like to highlight that this specific research, precisely, motivated by the research of Tonetti and other relevant researchers in the field, a raised by the reviewer, is now being carried out by the research team, to further explore these associations.

Point 3: The discussion should be limited more to strictly necessary elements, part of the contents of previous studies should be incorporated in the introduction. This will help shorten the length of the discussion. In the same way, the conclusions are excessively long, they must be delimited with the essential text (for example, the part developed in the first paragraph of page 11 can be summarized in a single sentence avoiding speculations of prevention).

Response 3:

The discussion has been revised and some sentences have been shortenned. Also, Conclusions have been modified, shortenning different sections as suggested by reviewer 2, and changing the reference to the last paragraph of the introduction:

“Taking into account these considerations, the objective of this study was to identify predictive factors during screening and assessments in a pediatric emergency service setting, to allow for differentiating of suicide attempts with an increased risk of re-attempt from those that do not carry that risk, with the ultimate goal of designing and prioritizing tailored interventions for these individuals, understanding that suicide is rarely due to a single cause and requires a range of prevention initiatives and evaluation methods (Al-Halabi & Fonseca-Pedrero, 2021)”.

Despite this we consider it’s important to clarify all the concepts as we did in the discussion giving to the lector an actual revision of the theme, in relaton to our main findings.

Point 4: The reference section should be checked carefully. There are numerous errors, such as the current references 1-2, 3-4, 5-6, 7-8, 9-10,... in which the second contains only the link to the first. The authors must then assess the number of references the manuscript really has (probably half). Do they correspond to all those included in the text? No numbering has been used in this one. All of these formatting issues need to be adjusted accordingly.

Response 4:

We have adjusted the format of the references correcting the mistakes.

Point 5: The age of the participants must be detailed in the abstract and check to always use adolescents. In the current version the last sentence of the abstract uses “young subjects”.

Response 5:

We have included the age of the participants in the abstract:

“A longitudinal 12-month follow-up design was carried out in a sample of 533 Spanish adolescents between 12 to 17 years old”

We have also change “young subjects” alongg the manuscripot by using “adolescents” or equivalent terms.

Point 6: Finally, in the title it is necessary to specify that the adolescent population is Spanish.

Response 6:

Following your recommendation we change the tittle for: “Predictors of Suicide Re-Attempt in a Spanish Adolescent Population After 12 Months’ Follow-Up”.

Round 2

Reviewer 2 Report

The authors have made all the changes I suggested in my review, having considerably improved the second version of the manuscript.